# Physical Activity and Health-Related Quality of Life of Patients on Hemodialysis with Comorbidities: A Cross-Sectional Study

**DOI:** 10.3390/ijerph19020811

**Published:** 2022-01-12

**Authors:** Yu-Hui Wu, Yu-Juei Hsu, Wen-Chii Tzeng

**Affiliations:** 1Graduate Institute of Medical Sciences, National Defense Medical Center, Taipei 11490, Taiwan; nana197926@mail.ndmctsgh.edu.tw; 2Nursing Department, Tri-Service General Hospital, Taipei 11490, Taiwan; 3Nephrology Division, Tri-Service General Hospital, Taipei 11490, Taiwan; yujuei@gmail.com; 4School of Medicine, National Defense Medical Center, Taipei 11490, Taiwan; 5School of Nursing, National Defense Medical Center, Taipei 11490, Taiwan

**Keywords:** hemodialysis, comorbidity, physical activity, health-related quality of life (HRQoL)

## Abstract

Patients on hemodialysis with multiple comorbidities have limited physical activity, resulting in poor health, low activity participation, and low quality of life. Accordingly, the nursing care provided to such patients should include regular physical activity training programs. Therefore, this cross-sectional descriptive study investigated whether patients on hemodialysis with and without comorbidities have different levels of physical activity and health-related quality of life (HRQoL); the correlations among the comorbidities, physical activity, and HRQoL of the two cohorts were also assessed. The 36-Item Short-Form Health Survey version 2 and International Physical Activity Questionnaire were employed to collect data from 120 patients on hemodialysis. An independent samples *t*-test and univariate and multivariate linear regression analyses were conducted. The overall HRQoL of patients with comorbidities was lower than that of patients without comorbidities (*p* = 0.008). Compared with patients who participated in low-intensity physical activity, the overall HRQoL of patients who participated in moderate-intensity physical activity was higher (*p* < 0.001). The overall HRQoL of patients with comorbidities who participated in low-intensity physical activity was lower than that of those who participated in moderate-intensity physical activity (*p* < 0.001). Moderate-intensity physical activity was correlated with higher HRQoL for patients with comorbidities. This finding supports the implementation of effective physical activity intervention measures. Furthermore, it supports the promotion of patient self-management and the implementation of regular exercise programs and lifestyle changes, and patients on hemodialysis can benefit from the future management of physical activities.

## 1. Introduction

The most common comorbidities of patients on hemodialysis are hypertension, diabetes, and cardiovascular diseases. These comorbidities can cause complications and can result in limited physical activity and low quality of life [1,2,3]. In addition, the risk of patients on hemodialysis with declining physical function and reduced muscle and bone mass is higher than that of normal people. Studies have suggested that patients should exercise at least five days a week and engage in moderate-intensity activities that last for 30 min or longer [4]. Their energy expenditure must reach at least 600 metabolic equivalents of tasks (METs). An energy expenditure of less than 600 METs among these patients indicates that their physical activity level is insufficient, and they should intensify their physical training to prevent further decline in their physical function [5,6].

In Taiwan, chronic kidney disease (CKD) is the ninth most common cause of death [7], and the incidence of hemodialysis increased from 10,668 people in 2014 to 12,346 in 2018. The prevalence of hemodialysis in Taiwan is 3587 per million people, and the number of patients on hemodialysis is increasing by 3% to 4% annually. Approximately 94,000 patients undergo hemodialysis every year. This figure is the highest globally, and hemodialysis is becoming a major public health problem in Taiwan [8]. Approximately 40% of patients on hemodialysis have two or more comorbidities [8]. The development of diseases and clinical abnormalities in these patients causes negative effects, such as the rapid reduction of muscle tension and strength, reduced physical activity, poor prognoses, and low health-related quality of life (HRQoL) [9,10].

Patients with CKD must undergo two to three sessions of hemodialysis every week, with each session lasting between 3 and 4 h. Therefore, compared with healthy people, their physical activity level is 35% lower, and their physical tolerance is also lower [11,12,13]. Approximately 47.4% of all patients have limited time for physical activity and become accustomed to a sedentary lifestyle. With time, patients on hemodialysis tend to develop symptoms, such as fatigue, muscle soreness, and cramping as well as reduced lower extremity muscle strength. The severity of these symptoms is negatively correlated with their physical activity levels and HRQoL. Relative to patients without multiple comorbidities, those with multiple comorbidities have poorer physical health and limited physical activity, resulting in lower physical activity participation [11,12,14,15,16] and a higher mortality rate [17]. Furthermore, the presence of multiple comorbidities aggravates their diseases, contributes to their medical burden, and increases their mortality risk [10]. Few studies have evaluated how the presence of comorbidities in patients on hemodialysis affects their level of physical activity and HRQoL. Therefore, the present study compared the physical activity levels of patients on hemodialysis with comorbidities and those without comorbidities and determined how their level of physical activity affects their HRQoL.

## 2. Materials and Methods

### 2.1. Design

This study is a descriptive cross-sectional study.

### 2.2. Participants

Patients on hemodialysis were recruited from the outpatient clinic of a medical center in Northern Taiwan between January and December 2020. In total, 120 patients completed the 36-Item Short-Form Health Survey version 2 (SF-36) and the abbreviated version of the International Physical Activity Questionnaire (IPAQ) used in Taiwan (Figure 1). The Charlson comorbidity index (CCI) comprises 19 diseases that are weighted on the basis of their association with mortality [18]. The present study used the CCI to measure the comorbidity severity of patients on hemodialysis. As selected by physicians, the study population included patients who regularly received hemodialysis for ≥3 months at a frequency of three times a week, with each session lasting ≥3 h; were ≥20 years old; were conscious; were able to communicate clearly in Chinese or Taiwanese; were literate; and were willing to participate in the present study after the purpose of the study was explained to them. We excluded patients with cognitive disabilities or mental illnesses (because such patients cannot properly respond to our questionnaire), patients who could not care for themselves, and patients who were hospitalized at the time of recruitment.

The required sample number was estimated using the G*Power version 3.1.9 [19] software. A linear multiple regression model F-test was conducted, with the effect size (f^2^), significance level, and power being 0.2, 0.05, and 0.80, respectively. With a projected attrition rate of 10%, the required sample size was estimated to be ≥104.

### 2.3. Measurement

#### 2.3.1. Demographics

The demographic data of the participants included age, gender, education, marital status, living arrangement, current employment, monthly income, body mass index (BMI, kg/m^2^), comorbidities, regular physical activity (three times/week), and duration of hemodialysis (years). The biochemical data of the participants included dialysis efficiency (Kt/V), normalized protein catabolic rate (nPCR), hemoglobin level (mg/dL), blood urea nitrogen level (BUN; mg/dL), creatinine level (mg/dL), and albumin level (g/dL) [20].

#### 2.3.2. HRQoL

HRQoL was assessed using the Medical Outcomes Study 36-Item Short-Form Health Survey version 2 (SF-36). The questionnaire contains 36 items under eight subscales as follows: physical functioning (10 items), role-physical (4 items), bodily pain (2 items), general health (5 items), vitality (4 items), social functioning (2 items), role-emotional (3 items), mental health (5 items), and health transition (1 item). The eight subscales belong to the two constructs of physical component score (PCS) and mental component score (MCS) [21]. The total score ranges from 0 to 100, with a higher score indicating better health and higher quality of life [22]. The Cronbach alpha of the questionnaire has been reported as being greater than 0.70 [22].

#### 2.3.3. Physical Activity Measure

The present study used Taiwan’s abbreviated version of the International Physical Activity Questionnaire (IPAQ) for measuring physical activity [23]. The scale was used to examine how much time the participants spent on physical activities over the past 7 days, the types of physical activities they engaged in that lasted more than 10 min, and the duration of such activities. Physical activity intensity was measured using physical activity metabolic equivalent of task (MET; kcal/h/kg). MET is the product of the resting metabolic rate multiplied by time [6]. The activities were categorized as low-intensity (<600 MET-min/week), moderate-intensity (600–2999 MET-min/week), and vigorous-intensity (≥3000 MET-min/week) activities, and their overall physical activity score was calculated. The IPAQ has excellent stability; Spearman’s rho was used to test the IPAQ, and the results indicated that the IPAQ had a reliability of 0.8 and criterion validity of 0.30 [24]. Therefore, the IPAQ is precise and effective.

### 2.4. Ethical Considerations

The present study enrolled participants after obtaining approval from the relevant institutional review board (IRB number: 1-108-05-195). The participants provided oral and written consent, and they had the right to exercise their autonomy and refuse participation in or withdraw from the study at any time; the medical rights of the participants were not affected by their participation in the present study. All of the data collected in the present study were coded and used only for research purposes.

### 2.5. Data Analysis

Statistical analysis was performed using SPSS version 22.0 (SPSS, Chicago, IL, USA), with the significance level set at 0.05. The participants’ sociodemographics, quality of life, and physical activity data are presented as means, standard deviations (SDs), and percentages (%). The associations among comorbidities, quality of life, and physical activity were evaluated using *t*-tests and generalized linear modeling (GLM). Multivariate analyses based on linear regression models were conducted, with adjustment for sociodemographic characteristics (age, gender, education, marital status, employment, monthly income, regular physical activity, duration of hemodialysis, and comorbidities) and physical activity.

## 3. Results

We enrolled 120 patients on hemodialysis who were aged between 24 and 84 years (mean age = 61.46 years) as participants in the present study. Most participants were male (77.5%), and 62.5% of the participants had a senior high school or higher level of education. Among the participants, 65.8% were married, 89.2% lived with their family, 81.7% did not have an occupation, and 90.0% had a personal income of <50,000 NTD. Their average hemodialysis time was 5.68 years (*SD* = 4.37). Among the participants, 49.2% had comorbidities that were common, namely congestive heart failure (34.2%), peripheral vascular disease (77.5%), and diabetes (47.5%); 50.8% performed regular physical activity; and only 45% reported sufficient energy expenditure (≥600 METs/week). Furthermore, the participants’ total mean score for HRQoL was 63.11 (*SD* = 16.20), their average PCS was 65.37 (*SD* = 17.55), and their average MCS was 60.85 (*SD* = 14.85; Table 1).

Table 2 presents the univariate linear regression analysis results. For the PCS construct, patients with >12 years of education (*B* = 10.58, *p* < 0.001, 95% confidence interval [CI]: 4.40–16.76), married (*B* = 6.57, *p* = 0.047, 95% CI: 0.08–13.06), current employment (*B* = 18.58, *p* < 0.001, 95% CI: 11.21–25.95), monthly income was >50,000 NT$ (*B* = 21.74, *p* < 0.001, 95% CI: 12.07–31.41), an increase in Kt/V (*B* = 10.86, *p* = 0.028, 95% CI: 1.16–20.56), a higher hemoglobin level (*B* = 2.69, *p* = 0.020, 95% CI: 0.42–4.95), regular weekly physical activity (*B* = 8.28, *p* = 0.008, 95% CI: 2.20–14.36), and weekly energy expenditure of ≥600 METs (*B* = 74.46, *p* < 0.001, 95% CI: 71.24–77.69) were positively correlated, and the presence of comorbidities (*B* = −8.30, *p* = 0.008, 95% CI: −14.56 to −2.15) was negatively correlated.

For the MCS construct, patients with >12 years of education (*B* = 6.76, *p* = 0.013, 95% CI: 1.43–12.10), current employment (*B* = 11.87, *p* < 0.001, 95% CI: 5.36–18.37), monthly income was >50,000 NT$ (*B* = 18.10, *p* < 0.001, 95% CI: 9.89–26.30), an increase in Kt/V (*B* = 11.51, *p* = 0.005, 95% CI: 3.40–19.62), regular weekly physical activity (*B* = 5.48, *p* = 0.039, 95% CI: 0.28–10.69), duration of hemodialysis (*B* = 0.70, *p* = 0.020, 95% CI: 1.11–3.66), and weekly energy expenditure of ≥600 METs (*B* = 67.92, *p* < 0.001, 95% CI: 64.86–70.97) were positively correlated, and the presence of comorbidities (*B* = −6.46, *p* = 0.015, 95% CI: −11.68 to −1.23) was negatively correlated.

Table 3 presents the result of the independent samples *t*-test, which revealed that the patients on hemodialysis with comorbidities had lower overall HRQoL (*t* = 2.70, *p* = 0.008), PCS (*t* = 2.83, *p* = 0.005), MCS (*t* = 2.29, *p* = 0.024), and weekly energy expenditure of ≥600 METs (*t* = 3.35, *p* < 0.001) than those without comorbidities. Multivariate analysis was conducted, with adjustment for the demographic statistics and factors of physical activity. Compared with the patients on hemodialysis without comorbidities, those with comorbidities had lower HRQoL but reported a comparable level of physical activity.

Table 4 presents the results of univariate and multivariate linear regression analyses. In the analyses with adjustment for age, gender, education, marital status, employment, monthly income, Kt/V, hemoglobin level, regular physical activity, habitual physical activity, duration of hemodialysis, and comorbidities, the HRQoL of the patients on hemodialysis who engaged in moderate-intensity physical activity was significantly higher than that of the patients who engaged in low-intensity physical activity (Table 4). In addition, among the patients with comorbidities, overall HRQoL (*p* = 0.039), PSC (*p* = 0.011), physical functioning (*p* = 0.011), and vitality (*p* = 0.023) were significantly higher among the patients who engaged in moderate-intensity physical activity than among those who engaged in low-intensity physical activity (Figure 2).

When comparing HRQoL between the patients with comorbidities who engaged in the low-intensity physical activity and moderate-intensity physical activity, those engaging in moderate-intensity physical activity had better HRQoL; particularly, they had higher PCS, physical functioning, role-physical, MCS, vitality, and social functioning.

## 4. Discussion

Our results revealed that the overall quality of life of patients on hemodialysis with comorbidities was poorer than that of patients on hemodialysis without comorbidities. In addition, the patients on hemodialysis with comorbidities who exercised regularly had higher quality of life than those who did not exercise regularly. That is, engagement in physical activity improved the physical functioning and spirit of these patients. This confirmed that in patients on hemodialysis, regular exercise is a feasible and effective activity that can improve HRQoL and muscle strength and help delay their disease progression [25,26]. For these patients, the implementation of moderate-intensity exercise training programs (e.g., aerobic exercises) can effectively improve the physical role, general health, and pain management domains of their HRQoL; increase their social interactions and enhance their self-attitude [27]; alleviate their disease-related conditions and complications; and improve their physical functioning and prognosis [28,29]. A study indicated that the level of physical activity of patients on hemodialysis is correlated to their HRQoL; the study also reported that relative to healthy people, habitual exercise is a stronger predictor of HRQoL for patients on hemodialysis [30]. Therefore, exercise and physical activity are crucial for treating and preventing multiple diseases. In other words, exercise is medicine [31].

Patients on hemodialysis usually have multiple comorbidities. Compared with patients with a single disease, the HRQoL and physical functioning of patients with multiple comorbidities are more likely to be affected by various factors [32]. Studies have demonstrated that comorbidities aggravated the primary disease of patients and affected their general health and MCS [33]. The presence of multiple comorbidities is correlated with reduced PCS, MCS, and overall HRQoL [2]. Multiple studies have confirmed that comorbidities and diseases affect each other and, consequently, the physical functioning and survival of patients. Comorbidities aggravate the severity of diseases; for patients, quality of life decreases as the number of comorbidities increases, and the physical health of these patients are affected to a greater degree than their mental health [34]. Compared with patients on hemodialysis without comorbidities, the overall HRQoL of patients on hemodialysis with comorbidities is lower, and they are also more likely to experience declining health [34]. Therefore, medical personnel should focus on improving the HRQoL of patients on hemodialysis with comorbidities.

Physical activity is beneficial for CKD [35]. Regular physical activity can improve the physical functioning, muscle tension and strength, PCS, and MCS of patients [25,36]. In patients on hemodialysis, performing regular moderate-intensity exercises three times per week, with each exercise session lasting for more than 30 min, can improve their PCS, MCS [37], and HRQoL in terms of the pain, physical role, and general health domains [24,38]. Exercise can also improve the physical functioning of patients on hemodialysis, stabilize their cognitive function, reduce their risk of developing health conditions, and improve their quality of life [39]. Regular exercise can improve the physical functioning and HRQoL of patients [40]. Patients on hemodialysis who engage in physical activity exhibit high quality of life [13]. The quality-of-life score and daily physical activity are positively correlated [41]. Hornik et al. [13] reported that compared with patients on hemodialysis who did not exercise regularly and engaged in less physical activity, patients who exercised regularly according to physical activity plans showed a lower incidence rate of complications; in addition, regular exercise improved their hemodialysis, physical functioning, role limitations of physical problems, social function, and PCS. The results of the present study revealed that the patients who engaged in moderate-intensity physical activity every week had higher HRQoL than the patients who did not engage in moderate-intensity physical activity every week. That is, patients on hemodialysis with low physical activity had low quality of life. Therefore, engaging in regular physical activity and increasing the level of physical activity can improve the HRQoL of patients on hemodialysis.

A recent study in Taiwan reported a correlation between comorbidities and the loss of HRQoL in analyses with adjustment for sociodemographic factors and medical comorbidities; this discovery is consistent with the results of the present study [42]. Complications and the physical activity level affect HRQoL, possibly because an increase in the number of comorbidities result in deteriorating health, limited physical activity, and reduced activity participation among patients on hemodialysis [34], thereby affecting their HRQoL. When patients engaged in more physical activity, their muscle strength and physical functioning improved; this also improved the waste removal ability of their kidneys and the efficiency of their hemodialysis. Consequently, their HRQoL improved [28,41]. Patients on hemodialysis with comorbidities can exercise regularly to improve their muscle strength and cardiovascular function, prevent cardiovascular diseases, and increase their survival rate, and benefits can also be gained from social interactions. In addition, regular exercise can improve their quality of life and physical functioning and can reduce the negative effects of complications [36,42]. Our results revealed the HRQoL of the patients on hemodialysis who engaged in moderate-intensity physical activity was lower than those without comorbidities. Therefore, we can actively encourage patients on hemodialysis with comorbidities to follow regular moderate-intensity physical activity training programs to improve their HRQoL, reduce the negative effects caused by their comorbidities, and enhance their overall health.

We also observed that regular physical activity and increased levels of physical activity are correlated with HRQoL. Moreover, the educational level, marital status, occupation, income, comorbidity, and regular hemodialysis of patients are associated with their HRQoL. However, similar to other studies, sufficient evidence was not provided in the present study to support the correlation of age and gender with HRQoL [43].

Studies have indicated that patients who have a lower educational level or no education, patients who are not married or widowed, patients who are unemployed, patients who have a low monthly income, and patients who have multiple comorbidities usually have poorer mental health, which directly affects their HRQoL and is negatively correlated with their quality of life. In addition, the quality of life of patients is negatively affected by a hemodialysis history of <2 years, lack of regular exercise, and lack of moderate-intensity exercise [29,43,44]. Therefore, regular physical activity is positively correlated with improvements in quality of life, PCS, and MCS. Physical activity is a predictor of quality of life [45]. Daily moderate-intensity physical activity is correlated with improved SF-36 scores (physical functioning, freedom from pain, vitality, and mental health) [46], and it reduces complications and improves physical functioning, HRQoL, and prognoses [27,28].

The present study has several limitations. First, the study participants were patients from a single hemodialysis center in Northern Taiwan. Therefore, our results may have limited generalizability. Second, the cross-sectional design precluded the determination of long-term changes in physical activity and HRQoL. Finally, the IPAQ used in the present study is a review scale and does not consider objective data. The other data in the scale were self-reported by the patients, who could have underestimated or overestimated their conditions; hence, the reliability and authenticity of the results are lower.

Our findings indicate that the HRQoL of patients on hemodialysis, particularly those with comorbidities, can be improved by providing early intervention involving regular moderate-intensity physical activity. Healthcare professionals should implement effective physical activity interventions to improve the physical activity level of these patients and encourage them perform regular physical activity and adopt a more active lifestyle; these changes will improve their quality of life.

## 5. Conclusions

The present study revealed that among the patients on hemodialysis who engaged in low-intensity physical activity, those with comorbidities had lower HRQoL relative to those without comorbidities. Furthermore, among the patients on hemodialysis with comorbidities, those who engaged in moderate-intensity physical activity had higher HRQoL than those who engaged in low-intensity physical activity; this was because the latter group had a sedentary lifestyle that resulted in disabilities, and comorbidities aggravate the decrease in muscle tension, strength, and physical activities, all of which led to a poor prognosis. In our cohort, the quality of life of patients with comorbidities who exercised regularly was higher than that of patients with comorbidities who did not exercise regularly. Our findings support the promotion of moderate-intensity physical activity training programs, which can improve quality of life. These programs should play an essential role in the treatment plans and health promotion measures for patients on long-term hemodialysis with comorbidities.

## Figures and Tables

**Figure 1 ijerph-19-00811-f001:**
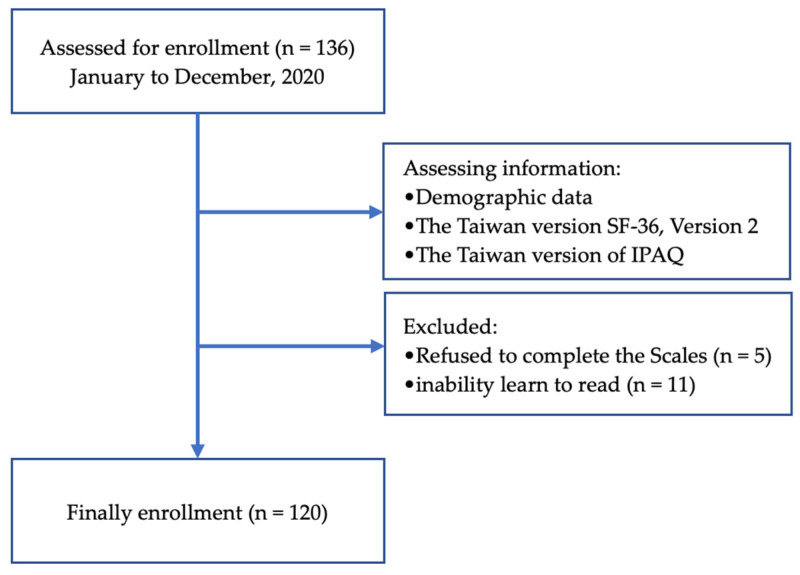
Flowchart for the present cross-sectional study.

**Figure 2 ijerph-19-00811-f002:**
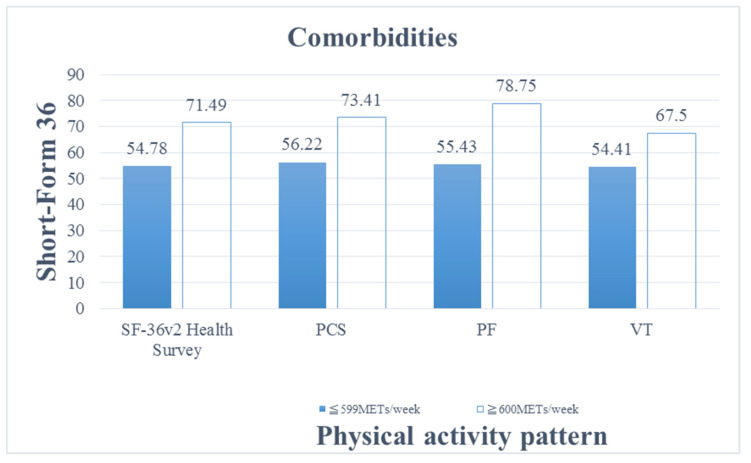
Comparison of physical activity patterns and health-related quality of life of patients on hemodialysis with comorbidities. Abbreviations: Physical Component Summary, PCS; physical functioning, PF; vitality, VT.

**Table 1 ijerph-19-00811-t001:** Sociodemographic characteristics of patients on hemodialysis (*n* = 120).

Variables	N	%	Mean ± SD
Age (years)			61.46
≤59	45	37.5	
≥60	75	62.5	
Gender			
Male	93	77.5	
Female	27	22.5	
Education			
elementary/Junior high School	45	37.5	
Senior high School or above	75	62.5	
Marital status			
Unmarried	41	34.2	
Married	79	65.8	
Living arrangement			
Alone	13	10.8	
with family	107	89.2	
Current employment			
Yes	22	18.3	
No	98	81.7	
Monthly income/(NTD)			
<50,000	108	90.0	
≥50,000	12	10.0	
BMI, kg/m^2^			23.74 ± 4.10
<25	79	65.8	
25–29.9	31	25.8	
≥30	10	8.4	
Biochemical data			
Kt/V			1.49 ± 0.31
nPCR (g/kg/d)			1.16 ± 0.27
Hemoglobin (mg/dL)			10.37 ± 1.43
Blood urea nitrogen (BUN; mg/dL)			65.22 ± 18.88
Creatinine (mg/dL)			9.73 ± 2.23
Albumin (g/dL)			3.92 ± 0.33
Comorbidities			
Yes	59	49.2	
No	61	50.8	
Regular Physical activity (3 times/week)			
Yes	61	50.8	
No	59	49.2	
Duration of hemodialysis (years)			5.68 ± 4.37
IPAQ (METs/week)			
≥600 METs/week	54	45	
≤599 METs/week	66	55	
SF-36v2 Health Survey (total mean score)			63.11 ± 16.20
Physical Component Summary (PCS)			65.37 ± 17.55
Physical Functioning (PF)			69.69 ± 21.72
Role-Physical (RP)			56.66 ± 31.86
Bodily Pain (BP)			83.10 ± 18.28
General Health (GH)			52.04 ± 17.49
Mental Component Summary (MCS)			60.85 ± 14.85
Vitality (VT)			59.83 ± 11.61
Social Functioning (SF)			65.31 ± 21.03
Role-Emotional (RE)			59.44 ± 29.74
Mental Health (MH)			58.83 ± 9.03

Abbreviations: NTD, New Taiwan Dollar; BMI, body mass index; IPAQ, International Physical Activity Questionnaire; Comorbidities, Charlson Comorbidity Index (CCI) of ≥3; nPCR, normalized protein catabolic rate.

**Table 2 ijerph-19-00811-t002:** Analysis of factors associated with health-related quality of life and sociodemographic characteristics (*n* = 120).

	SF-36v2 Health Survey									
	SF-36v2 Health Survey (PCS)					SF-36v2 Health Survey (MCS)				
		95% CI					95% CI			
Variables	B	Wald	Lower	Upper	*p*	B	Wald	Lower	Upper	*p*
Age	−0.32	1.18	−0.47	0.13	0.275	−0.28	0.46	−0.34	0.16	0.494
Gender male vs. female	−6.89	3.34	−14.27	0.49	0.067	−3.31	1.05	−9.62	3.00	0.300
Education <12 vs. ≥12 years	10.58	11.28	4.40	16.76	<0.001	6.76	6.19	1.43	12.10	0.013
Marital status single vs. married	6.57	3.94	0.08	13.06	0.047	5.35	3.63	−0.14	10.84	0.056
Current employment no vs. yes	18.58	24.44	11.21	25.95	<0.001	11.87	12.80	5.36	18.37	<0.001
Monthly income (NTD) <50,000 vs. ≥50,000	21.74	19.41	12.07	31.41	<0.001	18.10	18.68	9.89	26.30	<0.001
Biochemical data										
Kt/V	10.86	4.81	1.16	20.56	0.028	11.51	7.74	3.40	19.62	0.005
nPCR (g/kg/d)	8.58	2.69	−1.67	18.84	0.101	4.23	0.89	−4.51	12.97	0.343
Hemoglobin (mg/dL)	2.69	5.49	0.42	4.95	0.020	1.31	1.73	−0.64	3.26	0.188
Blood urea nitrogen (BUN; mg/dL)	−0.01	0.02	−0.17	0.15	0.882	−0.01	0.06	−0.15	0.12	0.064
Creatinine (mg/dL)	−1.18	2.51	−2.63	0.27	0.112	−0.31	0.24	−1.57	0.93	0.619
Albumin (g/dL)	7.31	2.45	1.83	16.46	0.118	6.51	2.78	1.14	14.16	0.095
Comorbidities (CCI) <3 vs. ≥3	−8.30	7.01	−14.56	−2.15	0.008	−6.46	5.86	−11.68	−1.23	0.015
Regular Physical activity (3 times/week) no vs. yes	8.28	7.13	2.20	14.36	0.008	5.48	4.27	0.28	10.69	0.039
Duration of hemodialysis (years)	0.64	3.19	0.94	3.87	0.074	0.70	5.38	1.11	3.66	0.020
IPAQ (METs/week) <600 vs. ≥600	74.46	2045.24	71.24	77.69	<0.001	67.92	1899.52	64.86	70.97	<0.001

Abbreviations: CI, confidence interval; PCS, physical component summary; MCS, mental component summary; CCI, Charlson Comorbidity Index. The *p* values were obtained using linear regression models.

**Table 3 ijerph-19-00811-t003:** Comparison of health-related quality of life and physical activity of patients with and without comorbidities (*n* = 120).

		Comorbidities				
	All	Yes	None			
Variables	(*n =* 120) Mean (SD)	(*n* = 59) Mean (SD)	(*n* = 61) Mean (SD)	*t*	*p*	*p^a^*
SF-36v2 Health Survey (total mean score)	63.11 (15.55)	58.87 (15.87)	66.25 (14.65)	2.70	0.008	<0.001
Physical Component Summary (PCS)	65.37 (17.55)	60.60 (17.62)	68.90 (16.76)	2.83	0.005	<0.001
Physical Functioning (PF)	69.69 (21.72)	62.23 (23.70)	75.21 (18.43)	4.20	<0.001	<0.001
Role-Physical (RP)	56.66 (31.86)	49.63 (32.02)	61.86 (30.95)	2.14	0.034	<0.001
Bodily Pain (BP)	83.10 (18.28)	82.40 (17.73)	83.62 (18.78)	0.52	0.719	<0.001
General Health (GH)	52.04 (17.49)	48.13 (18.62)	54.92 (16.14)	1.79	0.075	0.024
Mental Component Summary (MCS)	60.85 (14.85)	57.14 (15.81)	63.60 (13.57)	2.29	0.024	<0.001
Vitality (VT)	59.83 (11.61)	58.23 (12.28)	61.01 (11.03)	1.74	0.196	<0.001
Social Functioning (SF)	65.31 (21.03)	57.10 (22.53)	71.37 (17.69)	3.30	<0.001	<0.001
Role-Emotional (RE)	59.44 (29.74)	56.04 (31.18)	61.95 (28.60)	1.12	0.284	<0.001
Mental Health (MH)	58.83 (9.03)	57.17 (9.47)	60.05 (8.55)	1.52	0.129	0.002
IPAQ (METs/week)						
≥600 METs/week	1137.69 (1555.08)	673.35 (1206.05)	1586.80 (1724.27)	3.35	<0.001	0.329

Note: SD, standard deviation; *t*, independent *t*-test. The *p* values were obtained using univariate analysis based on linear regression models. The *p^a^* values were obtained using multivariate analysis based on linear regression models, with adjustment for sociodemographic characteristics (age, gender, education, marital status, employment, monthly income, Kt/V, hemoglobin level, regular physical activity, habitual physical activity, duration of hemodialysis, and comorbidities) and physical activity.

**Table 4 ijerph-19-00811-t004:** Regression of the association between physical activity patterns and health-related quality of life of patients on hemodialysis (*n* = 120).

	Total Physical Activity (*n* = 120)				
	<600 METs/week	≥600 METs/week			
Variables	(*n* = 66) Mean (SD)	(*n* = 54) Mean (SD)	*t*	*p*	*p^a^*
SF-36v2 Health Survey (total mean score)	56.50 (15.55)	71.19 (11.20)	−5.81	<0.001	<0.001
Physical Component Summary (PCS)	57.94 (17.82)	74.46 (12.21)	−5.79	<0.001	<0.001
Physical Functioning (PF)	59.83 (23.09)	81.75 (11.66)	−6.34	<0.001	<0.001
Role-Physical (RP)	44.88 (33.07)	71.06 (23.55)	−4.88	<0.001	<0.001
Bodily Pain (BP)	80.22 (19.24)	86.62 (16.53)	−1.92	0.056	<0.001
General Health (GH)	46.81 (17.37)	58.42 (15.53)	−3.81	<0.001	0.003
Mental Component Summary (MCS)	55.07 (14.82)	67.92 (11.55)	−5.20	<0.001	<0.001
Vitality (VT)	55.68 (11.82)	64.90 (9.13)	−4.69	<0.001	<0.001
Social Functioning (SF)	57.19 (19.98)	75.23 (17.92)	−5.15	<0.001	<0.001
Role-Emotional (RE)	50.63 (31.19)	70.21 (24.04)	−3.78	<0.001	<0.001
Mental Health (MH)	56.78 (9.72)	61.33 (7.45)	−2.82	0.006	<0.001

Note: SD, standard deviation; *t*, independent *t*-test. The *p* values were obtained using univariate analysis based on linear regression models. The *p^a^* values were obtained using multivariate analysis based on linear regression models, with adjustment for sociodemographic characteristics (age, gender, education, marital status, employment, monthly income, Kt/V, hemoglobin level, regular physical activity, habitual physical activity, duration of hemodialysis, and comorbidities), and physical activity.

## Data Availability

The datasets used and/or analyzed during the current study are available from the corresponding author on reasonable request.

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
