# Peer review of "Physical Activity and Health-Related Quality of Life of Patients on Hemodialysis with Comorbidities: A Cross-Sectional Study"

_ijerph, 2022, doi:10.3390/ijerph19020811_

Round 1
Reviewer 1 Report
Review for the manuscript entitled “Physical Activity and Health-Related Quality of Life of Patients on Hemodialysis with Comorbidities: A Cross-Sectional Study”.
This study analyzed a questionnaire survey to determine the impact of comorbidities on the quality of life (QOL) of dialysis patients. It was found that comorbidities have adverse effects on physical and mental status. The method of analysis is valid, and the results are not in dispute.
However, this reviewer has several concerns:
- The definition of comorbidities is not given in the Materials and Methods. The first line of the Introduction states, "The most common comorbidities of patients on hemodialysis are hypertension, diabetes, and cardiovascular diseases. From this sentence, does it mean that comorbidity was defined as hypertension, diabetes, and cardiovascular diseases (CVDs) in this study?
- If the "comorbidities" in this study are hypertension, diabetes, and CVDs, it seems unreasonable to examine the impact of CVDs on QOL while treating them in the same way as hypertension and diabetes. For example, there are professional athletes with diabetes and hypertension. Depending on the type of CVD (e.g., cerebral stroke and ischemic heart disease), the QOL of dialysis patients may be different.
- In Table 2, a variable called "gender" is listed. Does this mean male or female?
That’s all.
Author Response
"Please see the attachment."

Reviewer 2 Report
The authors demonstrated that the quality of life of patients on hemodialysis with comorbidities was worse than that of patients on hemodialysis without comorbidities. In addition, more active patients had better quality of life than those who had more sedentary lifestyle. Those findings are not novel and were repeatedly reported. It was previously found that there is a link between HRQOL, physical performance (measured also with pedometer), frailty and comorbidities. Recently, multicenter study was published DOI: 10.1038/s41598-021-00924-0 . Moreover, exercise programs for chronic kidney disease patients provided were beneficial and optimized functional capacity and quality of life in CKD patients. There are few points that need attention.
Introduction:
- It was mentioned that 9,4000 patients undergo hemodialysis every day. Is it correct?
- Low hemoglobin level is not a symptom.
Materials and Methods
- No data on laboratory values, dialysis performance.
- Comorbidities were not defined.
- Study flowchart is lacking.
Results:
- It would better to add CCI or other index of comorbidities.
- Any comorbidity in only 50 % of patients? Was hypertension included?
- Only univariate analysis of factors associated with HRQOL is reported.
- Data from Table 4 are repeated on Figure 1.
- In Table 4 it is comparison not correlation shown.
Finally, it cannot be concluded "patients on long-term hemodialysis with comorbidities have limited physical activity and thus declining physical functioning because their time is restricted by hemodialysis". It is cross-sectional study that does not show causality, only relation. The sentence "Health care professionals should provide intervention measures to patients with comorbidities such as moderate-intensity physical activity training programs and self-management programs to improve their quality of life." is not supported by study results. It could be included into discussion.
Author Response
"Please see the attachment."

Round 2
Reviewer 1 Report
The authors responded to my point and included an explanation of the Charlson comorbidity index (CCI) in the revision. This seems to be sufficient.
However, I felt that only Response 3 missed the point.
>Response 3: ” , we have indicated in Table 2 that “gender” means male or female (page 6)”
No, it is not.
My question means: Which gender tended to show a negative correlation (B: -6.89, p = 0.067) with the SF-36v2 Health Survey (PCS) in Table2? Male gender? Female gender?
Author Response
Point: The authors responded to my point and included an explanation of the Charlson comorbidity index (CCI) in the revision. This seems to be sufficient. However, I felt that only Response 3 missed the point. My question means: Which gender tended to show a negative correlation (B: -6.89, p = 0.067) with the SF-36v2 Health Survey (PCS) in Table2? Male gender? Female gender?
Response: Thank you for your encouragement. As suggested, we have presented the direction of the correlation between demographic variables and two SF-36 constructs. We have selected significant factors associated with PCS or MCS and summarized the results as follows.
For the PCS construct, patients with >12 years of education (B = 10.58, p < 0.001, 95% confidence interval [CI]: 4.40–16.76), married (B = 6.57, p = 0.047, 95% CI: 0.08–13.06), current employment (B = 18.58, p < 0.001, 95% CI: 11.21–25.95), monthly income was >50,000NT$ (B = 21.74, p < 0.001, 95% CI: 12.07–31.41), an increase in Kt/V (B = 10.86, p = 0.028, 95% CI: 1.16–20.56), a higher hemoglobin level (B = 2.69, p = 0.020, 95% CI: 0.42–4.95), regular weekly physical activity (B = 8.28, p = 0.008, 95% CI: 2.20–14.36), and weekly energy expenditure of ≥600 METs (B = 74.46, p < 0.001, 95% CI: 71.24–77.69) were positively correlated, and the presence of comorbidities (B = −8.30, p = 0.008, 95% CI: −14.56 to −2.15) was negatively correlated.
For the MCS construct, patients with >12 years of education (B = 6.76, p = 0.013, 95% CI: 1.43–12.10), current employment (B = 11.87, p < 0.001, 95% CI: 5.36–18.37), monthly income was >50,000NT$ (B = 18.10, p < 0.001, 95% CI: 9.89–26.30), an increase in Kt/V (B = 11.51, p = 0.005, 95% CI: 3.40–19.62), regular weekly physical activity (B = 5.48, p = 0.039, 95% CI: 0.28–10.69), duration of hemodialysis (B = 0.70, p = 0.020, 95% CI: 1.11–3.66), and weekly energy expenditure of ≥600 METs (B = 67.92, p < 0.001, 95% CI: 64.86–70.97) were positively correlated, and the presence of comorbidities (B = −6.46, p = 0.015, 95% CI: −11.68 to −1.23) was negatively correlated.”

Reviewer 2 Report
The authors corrected the manuscript according to my suggestions. The only remark is related to the description of table 4. In the text it is stated “table 4 presents the results of univariate and multivariate linear regression analyses” but in the description is „correlation between physical activity patterns and health-related quality of life of patients on hemodialysis (n= 120).
Author Response
Point: The authors corrected the manuscript according to my suggestions. The only remark is related to the description of table 4. In the text it is stated “table 4 presents the results of univariate and multivariate linear regression analyses” but in the description is correlation between physical activity patterns and health-related quality of life of patients on hemodialysis (n= 120).
Response: Thank you for your positive comments. We have revised the caption of table 4 as “Regression of the association between physical activity patterns and health-related quality of life of patients on hemodialysis (n = 120).”
